# Deep Generative Models for Generating Labeled Graphs

**Shuangfei Fan**
Virginia Tech
sophia23@vt.edu

**Bert Huang**
Virginia Tech
bhuang@vt.edu

## Abstract

As a new way to train generative models, *generative adversarial networks* (GANs) have achieved considerable success in image generation, and this framework has also recently been applied to data with graph structures. We identify the drawbacks of existing deep frameworks for generating graphs, and we propose labeled-graph generative adversarial networks (LGGAN) to train deep generative models for graph-structured data with node labels. We test the approach with different discriminative models as well as different GAN frameworks on various types of graph datasets, such as collections of citation networks and protein graphs. Experiment results show that our model can generate diverse labeled graphs that match the structural characteristics of the training data and outperforms all baselines in terms of quality, generality, and scalability.

## 1 Introduction

Graphs are powerful complex data structures that can describe collections of related objects. Such collections could be atoms forming molecular graphs, users connecting on online social networks, and papers connected by citations. The connected objects, or nodes, may be of different types or classes. Methods that reason about this flexible and rich representation can empower analyses of important, complex real-world phenomena. It is important to be able to learn generative models for the distributions of graphs, which can encode knowledge about the nature of the objects represented by graphs. In this paper, we introduce a method that learns generative models for labeled graphs in which the nodes and the graphs may have categorical labels.

Recently Goodfellow et al. (2014) described generative adversarial networks (GANs), which have been widely explored in computer vision and natural language processing (Zhang et al., 2017; Yu et al., 2017) for generating realistic images and text, as well as performing tasks such as style transfer. GANs are composed of two neural networks. The first is a generator network that learns to map from a latent space to the distribution of the target data, and the second is a discriminator network that tries to distinguish real data from candidates synthesized by the generator. Those two networks compete with each other during training and each improve based feedback from the other. The success of this general GAN framework has proven it to be a powerful tool for learning the distributions of complex data.

Motivated by the power of GANs, researchers have used it for generating graphs too. Bojchevski et al. (2018) proposed NetGAN, which uses the GAN framework to generate random walks on graphs. De Cao & Kipf (2018) proposed MolGAN, which generate molecular graphs using the combination of a GAN framework and a reinforcement learning objective. However, there are many limitations of existing methods, such as the generality to graphs with different structures and scalability to different sized graphs. Furthermore, they are unable to generate graphs with node labels, a critical feature of some graph-structured data.

The rapid development of deep learning techniques has also led to advances in representation learning in graphs. Many works have been proposed to use deep learning structures to extract high-level features from nodes and their neighborhoods to include both node and structure information (Kipf & Welling, 2017; Hamilton et al., 2017; Fan & Huang, 2017). These methods have been shown to be useful for many applications, such as link prediction and collective classification. Among all

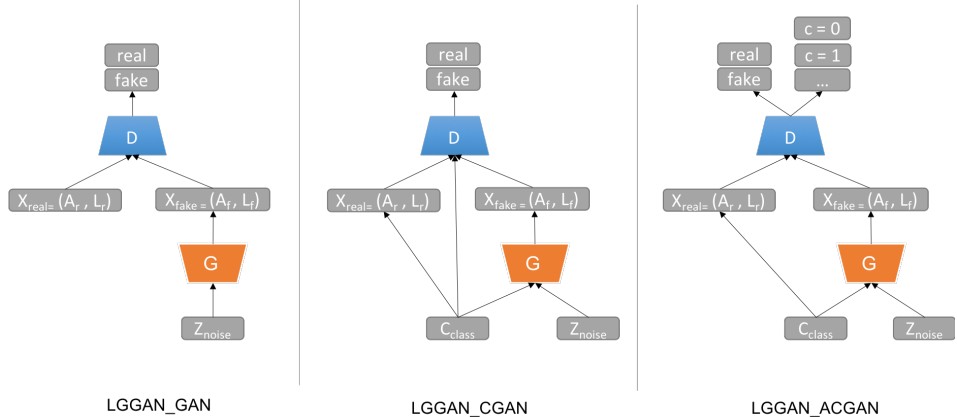

Figure 1: The adversarial training framework of LGGAN using different GAN structures. Conditional GANs and Auxiliary Conditional GANs incorporate increasing amounts of secondary information to aid the training process.

these works, Xu et al. (2018) propose jumping knowledge networks for learning adaptive, structure-aware representations that are particularly useful for representation learning on complex graphs with diverse graph structures.

Building on these advances, we propose *labeled graph generative adversarial network* (LGGAN), a deep generative model trained using a GAN framework to generate graph-structured data with node labels. LGGAN can be used to generate various kinds of graph-structured data, such as citation graphs, knowledge graphs, and protein graphs. Specifically, the generator in an LGGAN generates an adjacency matrix as well as labels for the nodes, and its discriminator uses a jumping knowledge network (Xu et al., 2018) to identify real graphs using adaptive, structure-aware higher-level graph features. Our approach is the first deep generative method that addresses the generation of labeled graph-structured data. In experiments, we demonstrate that our model can generate realistic graphs that preserve important properties from the training graphs. We evaluate our model on various datasets with different graph types—such as ego networks and proteins—and with different sizes. Our experiments demonstrate that LGGAN effectively learns distributions of different graph structures and that it can scale up to generate large graphs without losing much quality.

## 2 RELATED WORK

Generative graph models were pioneered by Erdös & Rényi (1959), who introduced random graphs where each possible edge appears with a fixed independent probability. More realistic models followed, such as the preferential attachment model of (Albert & Barabási, 2002), which grows graphs by adding nodes and connecting them to existing nodes with probability proportional to their current degrees. Goldenberg et al. (2010) proposed the stochastic block model (SBM), and Airoldi et al. (2008) proposed the mixed-membership stochastic block model (MMSB). The SBM is a more complex version of the Erdös-Rényi (E-R) model that can generate graphs with multiple communities. In SBMs, instead of assuming that each pair of nodes has identical probability to connect, they predefine the number of communities in the generated graph and have a probability matrix of connections among different types of nodes. Compared to the E-R model, SBMs are more useful since they can learn more nuanced distributions of graphs from data. However, SBMs are still limited in that they can only generate graphs with this kind of community structure.

With the recent development of deep learning, some works have proposed deep models to represent the distribution of graphs. Li et al. (2018) proposed DeepGMG, which introduced a framework based on graph neural networks. They generate graphs by expansion, adding new structures at each step. Li et al. (2018) proposed GraphRNN, which decomposes the graph generation into generating node and edge sequences from a hierarchical recurrent neural network. Simultaneously, researchers have also been developing other implicit yet powerful methods for generating graphs, especially based on the success of generative adversarial networks (Goodfellow et al., 2014). For example, Bojchevski et al.

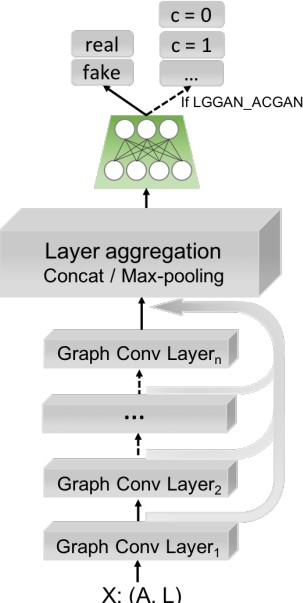

Figure 2: The structure of the discriminative model of LGGAN.

(2018) proposed NetGAN, which uses the GAN framework to generate random walks on graphs from which structure can be inferred, and De Cao & Kipf (2018) proposed MolGAN to generate molecular graphs using the combination of the GAN framework and reinforcement learning.

However, these recently proposed deep models are either limited to learning from a single graph (Kipf & Welling, 2016; Grover et al., 2018; Bojchevski et al., 2018), to generating small graphs with 40 or fewer nodes (Li et al., 2018), or to generating specific types of graphs such as molecular graphs (You et al., 2018a; De Cao & Kipf, 2018) (with no straightforward generalization to other domains due to specialized tools to calculate molecule-specific loss). Most broadly, most of these recently proposed methods cannot be used to generate labeled graphs.

## 3    MODEL

In this section, we introduce LGGAN and how it trains deep generative models for graph-structured data with node labels.

### 3.1    ARCHITECTURE

In this section, we provide details on the LGGAN architecture. As in the standard GAN framework, LGGAN consists of two main components: a generator $G$ and a discriminator $D$. The generator $G$ takes a sample from a prior distribution and generates a labeled graph $g$ represented by an adjacency matrix $A$ and a node label matrix $L$. The discriminator $D$ then trains to distinguish samples from the dataset and samples from the generator. In LGGAN, both the generator and the discriminator are trained using the improved Wasserstein GAN (Gulrajani et al., 2017) approach.

**Generator**    LGGAN's generative model uses a multi-layer perceptron (MLP) to produce the graph. The generator $G$ takes a random vector $z$ sampled from a standard normal distribution and outputs two matrices: (1) $L \in R^{N \times C}$, which is a one-hot vector that defines the node labels; and (2) the adjacency matrix $A \in R^{N \times N}$, which defines the connections among nodes in graphs. The architecture uses a fixed *maximum* number of nodes $N$, but it is capable of generating structures of fewer nodes.

Table 1: Comparison of LGGAN with different GAN frameworks and discriminative models on Cora-small and ENZYMES datasets.

| GAN frameworks | Cora_small | | | | ENZYMES | | | |
|---|---|---|---|---|---|---|---|---|
| | Degree | Clustering | Orbit | Label | Degree | Clustering | Orbit | Label |
| LGGAN_GAN_s | 0.27 | 0.18 | 0.03 | 0.37 | 0.67 | 0.88 | 0.004 | 0.01 |
| LGGAN_GAN | 0.21 | 0.14 | 0.007 | 0.15 | 0.31 | 0.20 | 0.01 | 0.008 |
| LGGAN_CGAN_s | 0.18 | 0.18 | 0.006 | 0.35 | 0.53 | 0.69 | 0.04 | 0.004 |
| LGGAN_CGAN | **0.10** | 0.24 | 0.01 | 0.19 | 0.23 | **0.13** | 0.02 | 0.01 |
| LGGAN_ACGAN_s | 0.14 | **0.009** | 0.06 | 0.13 | 0.51 | 0.29 | 0.03 | 0.01 |
| LGGAN_ACGAN | 0.13 | 0.08 | **0.03** | **0.11** | **0.09** | 0.17 | **0.005** | **0.01** |

**Discriminator** The discriminator $D$ takes a graph sample as input (represented by an adjacency matrix $A$ and a node label matrix $L$) and outputs a scalar value (and also a one-hot vector for class label if using the AC-GAN framework). For the discriminator, we propose two models, a simple one and an advanced discriminative model. The simple model is just an MLP while the advanced model uses the jumping knowledge networks (Xu et al., 2018) (JK-Net). The JK-Net is comprised by a series of graph convolution layers and a layer aggregation operator to integrate useful information from each layer for learning more powerful graph representations. We refer to the whole framework using the simple model as LGGAN_s and using the JK-Net as simply LGGAN. JK-Nets with layer aggregation have been shown to extract more adaptive and structure-aware representations compared to other representation learning methods such as graph convolutional networks (GCN) (Kipf & Welling, 2017), GraphSAGE (Hamilton et al., 2017) and graph attention networks (GAT) (Veličković et al., 2018). For our JK-Net, we use a GCN as the base model, and it propagates based on the following rule:

$$H^{(l+1)} = \sigma \left( \tilde{D}^{-\frac{1}{2}} \tilde{A} \tilde{D}^{-\frac{1}{2}} H^{(l)} W^{(l)} \right), \tag{1}$$

where $H(l) \in R^{N \times D}$ is output matrix at the $l-1$th layer, $I_N$ is the identity matrix, $\tilde{A} = A + I_N$ is the adjacency matrix of the graph $g$ with self-connections added, $\tilde{D}_{ii} = \sum_j \tilde{A}_{ij}$ is the diagonal degree matrix of graph $g$, $W^{(l)} \in R^{D \times F}$ is a the trainable weight matrix at $l$th layer, and $\sigma(\cdot)$ denotes an activation function (such as the sigmoid or ReLU (Nair & Hinton, 2010)). Since we do not include node attributes, we set $H(0) = I_N$ where $I_N$ is the identity matrix.

After $n$ layers of propagation via graph convolutions, we aggregate the outputs from each layer with an aggregation function agg, such as concatenation and max-pooling. We then concatenate the aggregated matrix with the node label matrix $L$ and outputs $Z_g$ as the final representation we learned for the graph $g$:

$$\begin{aligned} Z_g &= f(X, A, L) \\ &= \left[ \text{agg} \left( H^{(1)}, \ldots, H^{(n)} \right); L \right] \end{aligned} \tag{2}$$

The representation $Z_g$ of the graph will further be processed by a linear layer to produce the outputs of the discriminator: the graph-level scalar probability of the input being real data. The structure of this discriminative model is showing in Figure 2. If we are using the AC-GAN framework, there will also be a classifier to predict the category that the graph belongs to with a one-hot vector $c$.

## 3.2 ALTERNATE GAN FRAMEWORKS

Since the GAN framework was introduced by Goodfellow et al. (2014), many variations have been proposed that proved to be powerful for generation tasks. Therefore, we adopt three popular variations and compare how well they perform for our task of labeled graph generation. We use the traditional, original GAN approach as the first approach. Beyond the traditional GAN framework, we use two other methods that include extra information of classification labels for the graphs themselves. The first follows the *conditional GAN* (Mirza & Osindero, 2014) framework, which feeds the graph label as an extra input to the generator in addition to the noise $z$. We can use this label to generate the graphs of different types. To improve on this, our last variation uses the *auxiliary conditional GAN* (Odena et al., 2017) framework, in which the discriminator not only distinguishes whether the graph is real or fake, but it also incorporates a classifier of the graph labels. The structure of all those three variations is illustrated in Figure 1.

Table 2: Comparison of LGGAN and other generative models on different graph structured data using MMD evaluation metrics.

| | Cora_small | | | | Citeseer_small | | | | Cora | | | |
|---|---|---|---|---|---|---|---|---|---|---|---|---|
| | Degree | Clustering | Orbit | Label | Degree | Clustering | Orbit | Label | Degree | Clustering | Orbit | Label |
| E-R | 0.68 | 0.94 | 0.48 | N/A | 0.63 | 0.86 | 0.12 | N/A | 0.88 | 1.45 | 0.27 | N/A |
| B-A | 0.31 | 0.53 | 0.11 | N/A | 0.37 | 0.18 | 0.11 | N/A | 0.54 | 1.06 | 0.16 | N/A |
| MMSB | 0.21 | 0.68 | 0.07 | 0.48 | **0.17** | 0.50 | 0.11 | 0.32 | **0.12** | 0.68 | 0.09 | 0.49 |
| DeepGMG | 0.34 | 0.44 | 0.27 | N/A | 0.27 | 0.36 | 0.20 | N/A | - | - | - | - |
| GraphRNN | 0.26 | 0.38 | 0.39 | N/A | 0.19 | 0.20 | 0.39 | N/A | 0.20 | 0.46 | 0.11 | N/A |
| LGGAN | **0.13** | **0.08** | **0.03** | **0.11** | **0.17** | **0.13** | **0.04** | **0.09** | 0.15 | **0.21** | **0.06** | **0.009** |
| | Protein | | | | ENZYMES | | | | Citeseer | | | |
| | Degree | Clustering | Orbit | Label | Degree | Clustering | Orbit | Label | Degree | Clustering | Orbit | Label |
| E-R | 0.31 | 1.06 | 0.28 | N/A | 0.38 | 1.26 | 0.08 | N/A | 0.82 | 1.57 | **0.06** | N/A |
| B-A | 0.93 | 0.88 | 0.05 | N/A | 1.17 | 1.08 | 0.51 | N/A | 0.32 | 1.04 | 0.08 | N/A |
| MMSB | 0.46 | 1.05 | 0.21 | 0.01 | 0.55 | 1.08 | 0.05 | 0.92 | 0.08 | 0.50 | 0.11 | 0.32 |
| DeepGMG | 0.96 | 0.63 | 0.16 | N/A | 0.43 | 0.38 | 0.08 | N/A | - | - | - | - |
| GraphRNN | **0.04** | 0.18 | 0.06 | N/A | **0.06** | 0.20 | 0.07 | N/A | **0.20** | 1.15 | 0.14 | N/A |
| LG-GAN | 0.18 | **0.15** | **0.02** | **0.005** | 0.09 | **0.17** | **0.03** | **0.01** | 0.25 | **0.12** | **0.06** | **0.15** |

## 3.3 TRAINING

GANs (Goodfellow et al., 2014) train via a min-max game with two players competing to improve themselves. In theory, the method converges when it reaches a Nash equilibrium, where the samples produced by the generator matches the data distribution. However, this process is highly unstable and often results in problems such as mode collapse (Goodfellow, 2016). To deal with the most common problems in training GAN, such as mode collapse and unstable training, we use the Wasserstein GAN (Salimans et al., 2016) with a gradient penalty. We also adopt several techniques such as feature matching and minibatch discrimination that were shown to encourage convergence and help avoid mode collapse.

## 3.4 NODE ORDERING

A common representation for graph structure uses adjacency matrices. However, using matrices to train a generative model introduces the issue of how to define the node ordering in the adjacency matrix. There are $n!$ permutations of $n$ nodes, and it is time consuming to train over all of them.

For LGGAN, we use the framework of JK-Net with graph convolutions (Kipf & Welling, 2017) and a node aggregation operator (De Cao & Kipf, 2018) as the discriminator. This discriminator is invariant to node ordering, avoiding the issue. However our simplified version, LGGAN_s, uses a MLP as the discriminator, which is dependent on node ordering. Therefore, in this case, we arrange the nodes in a breadth-first-search (BFS) ordering for the training data. We adopt this strategy from You et al. (2018b), who generated graphs by adding nodes in a BFS ordering.

In particular, we preprocess the adjacency matrix $A$ and node label matrix $L$ by feeding them into a BFS function. This function takes a random permutation $\pi_g$ of the nodes in graph $g$ as input, picks a node $v_i$ as the starting node, and then outputs another permutation $\pi'_g$ that is a BFS ordering of the node in graph $g$ starting from node $v_i$. In this way, by specifying the node ordering in the graph, we only need to train on all possible BFS orderings, rather than all possible node permutations. This reduction makes a huge difference for computational complexity when graphs are large.

## 4 EXPERIMENTS

In this section, we first explore the six different variations of LGGAN (three different GAN frameworks for either LGGAN or LGGAN_s) and discuss their advantages and disadvantages. We also compare LGGAN with other graph generation methods to demonstrate its ability to generate high-quality labeled graphs in diverse settings.

## 4.1 BASELINES

We compare our model against various traditional generative models for graphs, as well as some recently proposed deep graph generative models. For traditional baselines, we compare against the Erdös-Rényi model (E-R) (Erdös & Rényi, 1959), the Barabási-Albert (B-A) model (Albert & Barabási, 2002), and mixed-membership stochastic block models (MMSB) (Airoldi et al., 2008). Then we also compare with some recently proposed deep graph generative models such as the Deep-GMG (Li et al., 2018) and GraphRNN (You et al., 2018b). Few current approaches are designed to generate labeled graphs. One exception is MolGAN (De Cao & Kipf, 2018), which is designed to generate molecular graphs and needs specialized evaluation methods specific to that task, so we do not compare against it. We also do not compare with NetGAN (Bojchevski et al., 2018) since its framework is constrained to learn to generate a single new graph from a single training graph.

## 4.2 DATASETS

We perform experiments on different kinds of datasets with varying sizes and characteristics, described in the following.

**Citation graphs**   We test on scientific citation networks. We used the Cora and Citeseer datasets (Sen et al., 2008). The Cora dataset is a collection of 2,708 machine learning publications categorized into seven classes, and the CiteSeer dataset is a collection of 3,312 research publications crawled from the CiteSeer repository. To test the scalability of LGGAN, we extracted different subsets with different graph sizes by constraining the number of nodes in graph $|V|$. For small datasets (denoted cora_small and citeseer_small), we extract two-hop and three-hop ego networks with $30 \leq |V| \leq 50$. For the large datasets (denoted cora and citeseer), we extract three-hop ego networks with $150 \leq |V| \leq 200$. When using the AC-GAN framework, we set the graph label to be the node label of the center node of the ego network.

**Protein graphs**   We also test on multiple collections of protein molecular graphs. The ENZYMES dataset consists of 600 enzymes (Schomburg et al., 2004). Each enzyme in the dataset is labeled with one of the six enzyme commission (EC) code top-level classes. The protein dataset includes proteins from the dataset of enzymes and non-enzymes created by Dobson & Doig (2003). There are two graph labels: enzymes and non-enzymes.

## 4.3 EVALUATION

To evaluate the quality of the generated graphs, we follow the approach used by You et al. (2018b): we compare a distribution of generated graphs with that of real ones by measuring the *maximum mean discrepancy* (MMD) (Gretton et al., 2012) of graph statistics, capturing how close their distributions are. We use four graph statistics to evaluate the generated graphs: degree distribution, clustering coefficient distribution, node-label distribution, and average orbit count statistics.

Since we are generating labeled graphs, we also want to evaluate the graph distribution in each class. To do this, we extract subgraphs for each class from both the training graphs and generated graphs and evaluate based on these three metrics (excluding the label distribution). These per-label tests help test whether the model simply assigns the class based on the label distribution without considering the underlying graph structure. Since these label-based evaluation metrics are only applicable to labeled graphs, we only apply them to MMSB and LGGAN.

## 4.4 COMPARING DIFFERENT VARIATIONS OF LGGAN

LGGAN is a flexible framework that can adopt different settings for each part of the training setup, such as the GAN framework and the model for generator and discriminator. The GAN variations we consider are (1) LGGAN_GAN, which uses the original GAN framework (Goodfellow et al., 2014), (2) LGGAN_CGAN, which uses the conditional GAN framework (Mirza & Osindero, 2014), and (3) LGGAN_ACGAN, which uses the AC-GAN (Odena et al., 2017) framework. We also compare two different discriminative models, a simple multi-layered perceptron and jumping knowledge networks (JK-Net) based on multiple layers of GCNs.

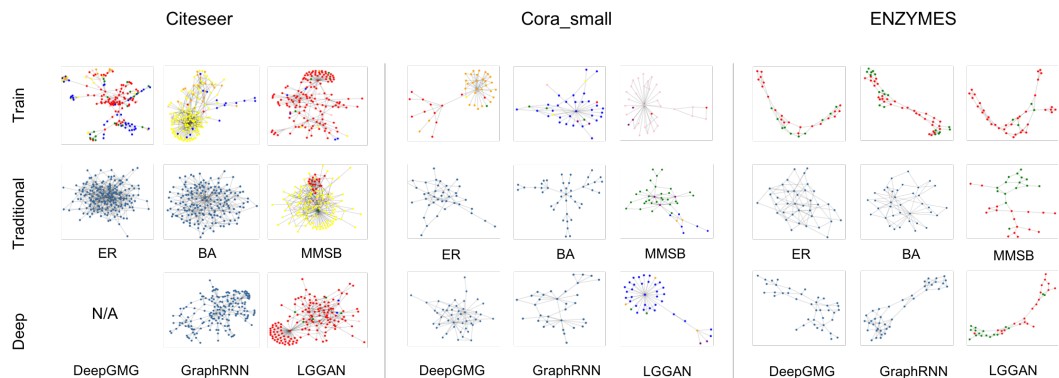

Figure 3: Visualization of training graphs (first row), graphs generated by traditional models (second row): E-R model, B-A model, MMSB model and graphs generated by deep models (third row): DeepGMG, GraphRNN, LGGAN for different datasets.

We evaluate these six possible setups for LGGAN on different graph-structured data to measure the quality of the learned generative models. We run experiments on two datasets: cora_small and EN-ZYMES. The results are listed in Table 1. Among all the three GAN frameworks, LGGAN_ACGAN achieves the best results on both datasets regardless of which discriminative model is used. This result matches with our expectations, since the AC-GAN framework incorporates the class information allowing it to learn a better embedding and to propagate that information to the generator.

For different discriminative models, using JK-Nets can improve the quality of generated graphs regardless of which GAN framework is used. An interesting point is that there is a gap between LGGAN_s and LGGAN on the ENZYMES dataset that is much larger than on the citation networks. This difference may be because the cora_small dataset is composed of many small two-hop and three-hop ego networks where the structure is quite simple and uniform—so it could be easier to learn. However, with the ENZYMES dataset, the structure is more complicated and diverse. Therefore it reveals that the LGGAN_s is unable to generalize to complex data. In contrast, the quality of generated graphs with LGGAN using JK-Net is more consistent among different datasets, which suggests that LGGAN can adaptively adjust to different graph-structured data. Based on these results, we use LGGAN_ACGAN in the remaining experiments and refer to it as LGGAN when comparing with other baselines.

## 4.5   COMPARING WITH OTHER METHODS

We compare LGGAN to other methods for generating graphs—both traditional generative models such as E-R, B-A, and MMSB as well as deep generative models that were proposed recently, such as GraphRNN and DeepGMG. DeepGMG cannot be used to generate large graphs due to its high computational complexity, so the results of DeepGMG on large graph datasets are not available. For each method, we measure three aspects. The first is the quality of the generated graphs, which should be able to mimic typical topology of the training graphs. The second is the generality, where a good generative model should be able to generalize to different and complex graph-structured data. Then the last aspect is the scalability, where we want the model to be able to scale up to generate large networks instead of being restricted to relatively small graphs.

Table 2 lists results from our comparison. LGGAN achieves the best performance on all datasets, with 90% decrease of MMD on average compared with traditional baselines, and 30% decrease of MMD compared with the state-of-the-art deep learning baseline GraphRNN. Although GraphRNN performs well on the two smaller protein-related datasets, ENZYMES and protein, it does not maintain the same performance on large datasets, such as cora and citeseer.

**Scalability**   To evaluate the scalability of these methods, we perform experiments on two different subsets of the Cora dataset with different graph sizes: the cora_small and cora datasets. As listed in Table 2, the traditional models all create a large gap between these two datasets in terms of three evaluation metrics. For the deep generative models, DeepGMG cannot be used to generate large

Table 3: Comparison of LGGAN with the other labeled graph generation model MMSB on both the graph statistics and average sub-graphs statistics of different classes using MMD evaluation metrics on ENZYMES dataset.

| | Graph statistics | | | | Sub-graph statistics | | |
|---|---|---|---|---|---|---|---|
| | **D**egree | **C**lustering | **O**rbit | **L**abel | Avg. **D** | Avg. **C** | Avg. **O** |
| MMSB | 0.55 | 1.08 | 0.05 | 0.92 | 0.14 | 0.20 | 0.03 |
| LGGAN | **0.09** | **0.17** | **0.03** | **0.01** | **0.13** | **0.15** | **0.01** |

graphs due to the computational complexity of it's generation procedure which try to add node one by one increasingly. And compared to GraphRNN, LGGAN MMD scores barely increase compare to the smaller dataset, suggesting that our model is more reliable and has the best ability to scale up to large graphs.

**Generality**   To evaluate the ability of LGGAN to adapt to different graph-structured data, we evaluate the results of all methods on the different domains of citation ego-networks (Cora) and molecular protein graphs (ENZYMES). From Table 2, LGGAN achieves more consistent results on various datasets compared to other models, where some of them suffer from the issue of generalization such as MolGAN and NetGAN which can only be used to generate specific or limited types of graph-structured data.

Some examples are visualized in Figure 3, which contains graphs generated by our model and the baselines. Although it is not as intuitive for humans to assess as, e.g., natural images, one can still see that LGGAN appears to capture the typical structures of datasets better than other models.

### 4.6   Evaluating Labeled Graphs

To better evaluate the structure of the labeled graphs being generated, we also calculate MMD of the three graph statistics for the sub-graphs centered around each node class, taking the average MMD value across all classes. Since among existing methods, only MMSB can be used to directly generate labels, we compare LGGAN to it using the ENZYMES datasets. The results are listed in Table 3. LGGAN can not only learn a good distribution of the labels, but it is also able to learn the structure within each class much more reliably than the MMSB model.

## 5   Conclusion

In this work, we proposed a deep generative model using a GAN framework that generates labeled graphs. These labeled graphs can mimic distributions of citation graphs, knowledge graphs, social networks, and more. We also introduced an evaluation method for labeled graphs to measure how well the model learns the sub-structure of the labeled graphs. Our model can be useful for simulation studies, especially when access to labeled graph data is limited by access or privacy concerns. We can use these models to generate synthetic datasets or augment existing datasets, to do graph-based analyses such as communication segmentation, node classification, anomaly detection, and link prediction. The experiments show that it outperforms other state-of-the-art models for generating graphs while also being capable of the previously unaddressed task of generating labels for nodes.

## 6   Acknowledgement

The authors would like to thank NVIDIA for donating hardware and Amazon for donating cloud computing credits to our group, we appreciate their support to our research.

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
