# OpenReview forum: "Deep Generative Models for Generating Labeled Graphs"
_ICLR.cc/2019/Workshop/DeepGenStruct — DeepGenStruct 2019_

### Official Review · AnonReviewer1 · 2019-04-15

**Rating:** 3
**Confidence:** 3

**Review:**

This paper introduces a new GAN model (LGGAN) for labeled graph generation. The output of LGGAN generator contains two parts: labels of each node (represented by one-hot vectors) and the adjacency matrix. The discriminator is a graph convolutional network that outputs graph-level scalar probability of the input being real data. Specifically, the author uses JK-Net to parameterize the discriminator. Empirical results show that LGGAN outperforms state-of-the-art baselines such as DeepGMG and GraphRNN in terms of the MMD evaluation metrics.

The proposed LGGAN model is very similar to MolGAN. In both model, the generator outputs the adjacency matrix and node label, and the discriminator is parameterized as GCN. The major difference is in the architectural choices of discriminator (JK-Net v.s. Relational GCN). Unfortunately the paper does not compare with MolGAN. It should be very easy to adapt MolGAN model for the datasets used in this paper. The reviewer is also concerned why LGGAN is not compared on molecule tasks. The "specialized evaluation method" has been established by previous work (e.g., MolGAN) and it's not hard to run at all.

Moreover, I have several important questions that needs to be clarified:
1) Since node labels and adjacency matrix are discrete values, how did the LGGAN propagates the gradient from the discriminator to the generator?
2) The author mentioned in Section 3.1 that the node attributes are not included in the discriminator. If so, the discriminator only focus on the adjacency matrix and the node labels would be mostly random. How would LGGAN learns the distribution of labeled graphs like Protein?

---

### Official Review · AnonReviewer2 · 2019-04-18
**GAN-based model for labeled graph generation**

**Rating:** 3
**Confidence:** 1

**Review:**

This paper proposes a GAN-based model, i.e., LGGAN to generate labeled graphs. The generator of LGGAN is a multi-layer perceptron, which samples from a standard normal distribution to generate the label matrix L and the adjacent matrix A. On the other hand, the discriminator takes a graph sample as input and outputs a scalar.

Empirical studies are conducted for generating citation and protein graphs. Under the maximum mean discrepancy (MMD) metric, LGGAN outperforms existing methods like MMSB and DeepGMG on both citation and protein graph generation. LGGAN also demonstrates advantages over MMSB in terms of the graph statistics, which is used to measure dissimilarity between the generated graph and the training graph.

Overall, the paper is well-written, the methodology makes sense to me, and the experimental results look good too. So I will vote for acceptance.

---

### Decision · Program_Chairs · 2019-04-19
**Acceptance Decision**

**Decision:**

Accept

**Comment:**

Accepted